Relationships among COVID-19 causal factors perceived by children, basic psychological needs and social anxiety

http://orcid.org/0000-0002-9921-744X González-García Higinio 1
http://orcid.org/0000-0003-0510-1161 Álvarez-Kurogi Leandro 2
http://orcid.org/0000-0002-2981-0782 Prieto Andreu Joel 1
http://orcid.org/0000-0001-8045-1693 Tierno Cordón Javier 1 javier.tierno@unir.net
Castro López Rosario 1
http://orcid.org/0000-0002-0830-8742 Salas Sánchez Jesús 1 3
1 Didactics of Physical Education and Health/Faculty of Education, Universidad Internacional de La Rioja (UNIR) , Logroño, La Rioja , Spain
2 Didactics of Physical Education and Health/Faculties of Education and Health Sciences, Universidad Internacional de La Rioja (UNIR) , Logroño, La Rioja , Spain
3 Universidad Autónoma de Chile , Temuco , Chile
Neiva Henrique
Electronic publication date: 2025 Jan 31
Publication date: 2025
Volume: 13
Electronic Location ID: e18828
Received 2024 Feb 23; Accepted 2024 Dec 17
Copyright: © 2025 González-García et al.
Copyright year: 2025
Copyright holder: González-García et al.
License: This is an open access article distributed under the terms of the Creative Commons Attribution License, which permits unrestricted use, distribution, reproduction and adaptation in any medium and for any purpose provided that it is properly attributed. For attribution, the original author(s), title, publication source (PeerJ) and either DOI or URL of the article must be cited.
License URL: https://creativecommons.org/licenses/by/4.0/

Keywords: PLS-PM, Epidemic, Perception, Children

Funding: Universidad Internacional de La Rioja This work was supported by the Universidad Internacional de La Rioja call for funding. The funders had no role in study design, data collection and analysis, decision to publish, or preparation of the manuscript.

==============================
Background

The pandemic caused by COVID-19 had a great impact on our society as the lives of children have been affected, as well as their psychological health and social anxiety.

Objective

To examine whether COVID-19 causal factors perceived by children predicted basic psychological needs and social anxiety, and if basic psychological needs predicted social anxiety.

Methods

A sample of 58 schoolchildren (Mage = 10.18; SD = 0.77; 36 boys, 22 girls) participated in the study and completed a series of self-report measures. The relationship between the study variables was examined using partial least square path modelling (PLS-PM).

Results

Social distancing and protection were found to significantly reduce competence satisfaction. Perceived psychological impact significantly negatively predicted relatedness satisfaction and significantly positively autonomy frustration and competence frustration. Perceived psychological impact significantly negatively predicted anxiety in the interaction with the opposite sex. Autonomy satisfaction significantly positively predicted anxiety in the interaction with the opposite sex. Autonomy frustration significantly positively predicted anxiety in public speaking interaction with teachers, significantly positively anxiety in the interaction with the opposite sex, significantly positively anxiety of being embarrassed or ridiculed and significantly positively anxiety in the interaction with strangers.

Conclusion

Perceived psychological impact was the causal factor that revealed a higher impact on basic psychological needs. As such, it is important to take measures with children in pandemic situations to minimize this variable. On the other hand, Autonomy frustration revealed a higher impact on social anxiety. Thus, it is necessary to emphasize autonomy in children respecting the restrictions imposed to minimize the impact of social anxiety.

Introduction

The COVID-19 epidemic has caused a large impact on society in distinct health aspects as well as social parameters (Ding et al., 2020; González-García, Fuentes & Renobell, 2022). As such, it is especially important to determine those causal factors perceived by children regarding the epidemic (Araújo et al., 2021).

Several authors have examined those COVID-19 causal factors perceived by the population (Akwa, Rene & Maingi, 2020; Conway, Woodard & Zubrod, 2020; Olapegba et al., 2020; Prieto, 2022). In particular, the COVID-19 causal factors are understood as the factors that the population considers or perceives as causing COVID-19 infection. In this study, it is followed the classification of perceived causal factors divided into (Prieto, 2022): Social Distancing and Protection (DSP), Perceived Psychological Impact (IPP), Skepticism (ES) and Credibility of Perceived Information (CIP). In particular, Social Distancing and Protection are those elements that the population perceives regarding maintaining a safe distance and the use or not of a mask as a protective element (Prieto, 2022). Perceived Psychological Impact is the stress and depression that the pandemic has generated and how the population relates this negative psychological impact as a causal risk factor for contagion (Prieto, 2022). Skepticism describes the skepticism and risk perception assumed by the population in situations involving restrictions, personal actions or social distancing as contagion risk prevention measures (Prieto, 2022). Credibility of information describes the insecurity and the credibility of the population towards the content related to COVID-19 transmitted by the government and by the media (Prieto, 2022). In this study the causal factors related to the epidemic are measured in schoolchildren which may reveal the influence of those causal factors in variables related to wellbeing such as basic psychological needs and social anxiety.

Another factor to take into account that was influenced during the pandemic of COVID-19, is the satisfaction of basic psychological needs (Cantarero, van Tilburg & Smoktunowicz, 2021; Dursun et al., 2022). According to Ryan & Deci (2000), Basic Psychological Needs are an intrinsic, universal, and necessary psychological requirement for psychological well-being. Basic psychological needs come from the Theory of Self-Determination (Ryan & Deci, 2000) and it is divided into the following components: autonomy (e.g., the necessity to decide about one’s actions, express ideas and opinions freely), competence (e.g., the necessity to feel ability when performing tasks, the experience of success and develop new skills), and relatedness (e.g., the importance of the close ties in the social relationships made) (González-Cutre et al., 2015; Ryan & Deci, 2000). In addition, another concept that is measured in this work is the deprivation of those aforementioned needs (Adie, Duda & Ntoumanis, 2008). This deprivation of needs is known as needs thwarting or frustration and it may turn into the compensation of them, which could be manifested by distinct unexpected behaviours (Kasser et al., 1995). For instance, a person with the frustration of the competence need may compensate it by pursuing image-oriented outcomes. Thus, the thwarting of the three basic psychological needs (competence frustration, autonomy frustration and relatedness frustration) will be examined in this work.

Basic psychological needs may be modified by the causal factors of the epidemic in different senses (Akwa, Rene & Maingi, 2020; Conway, Woodard & Zubrod, 2020; Olapegba et al., 2020; Prieto, 2022). In particular, social distancing may modify the perception of the relationship between people as well as the perception of people’s autonomy (Costa et al., 2022). On the other hand, skepticism and the credibility of the information may modify the behaviour patterns of people in complying with the rules and it may have an impact on basic psychological needs (Szántó & Dudás, 2022). The degree of perceived psychological impact can influence the fulfilment of basic psychological needs, depending on the behaviours people adopt (Šakan, Žuljević & Rokvić, 2020). This means that if the epidemic has significantly altered behaviour patterns, a person will experience greater frustration with their needs compared to cases where the epidemic has only slightly changed people’s behaviour patterns (Cantarero, van Tilburg & Smoktunowicz, 2021).

Another important factor, which has been modified after the epidemic, is the social anxiety disorder or social phobia (Zheng et al., 2020). From a theoretical point of view, social anxiety is understood as the consistent fear of one or more social situations or public performances in which there is an evaluation/observation to the person. This evaluation may imply a negative outcome that could display embarrassing or humiliating symptoms of anxiety (e.g., facial flushing, trembling, tics, etc.,) (American Psychiatric Association, 2013). In particular, previous works showed that COVID-19 increased the social anxiety of children (Gabarrell-Pascuet et al., 2021; Khan et al., 2021; Zheng et al., 2020). According to basic psychological needs, it makes sense that the frustration of relatedness may enhance the social anxiety experienced by children (Arad, Shamai-Leshem & Bar-Haim, 2021; Hawes et al., 2021). In this sense, the present research follows the social anxiety mode (Caballo et al., 2010; Caballo et al., 2012) This model divides social anxiety into the following factors: interaction with the opposite sex (SEX, e.g., anxiety when interacting with the opposite sex), public speaking/interaction with teachers (HABPUB, e.g., fear of public speaking or interacting with teachers), being embarrassed or ridiculed (EVI, e.g., fear of being embarrassed or ridiculed publicly), assertive expression of annoyance or anger (ASERT, e.g., difficulty expressing anger or annoyance assertively), interaction with strangers (DESCON, e.g., anxiety when interacting with unfamiliar people), and acting in public (ACT, e.g., discomfort performing actions in front of an audience).

However, little is known about how the perceived causal factors of COVID-19 increase the social anxiety of children and how basic psychological needs influence social anxiety. Nevertheless, following another related subject, it may be hypothesized that social distancing may decrease social anxiety (Arad, Shamai-Leshem & Bar-Haim, 2021), meanwhile, perceived psychological impact may be related to more social anxiety in children (Hawes et al., 2021). The person that experiences social anxiety tends to avoid the situations that he/she fears and if he/she cannot avoid them, he/she tries to escape or endures them with high discomfort (Caballo et al., 2011).

As a novelty, the present work is intended to examine the perceived causal factors of COVID-19 (Prieto, 2022) and their relationship with basic psychological needs and social anxiety. Those COVID-19 perceived causal factors may modify the well-being of children by impacting their social anxiety and basic psychological needs. The examination of those variables may help to reduce the epidemic outcomes and their negative effects on social anxiety and basic psychological needs. As such, the goals of the present study were to examine whether COVID-19 causal factors perceived by children predicted basic psychological needs and social anxiety; and whether basic psychological needs predicted social anxiety. The hypotheses established were: (a) Perceived Psychological Impact may have a positive relation with the frustration of basic psychological needs (competence, autonomy and relatedness) in children; (b) Skepticism may decrease the satisfaction of basic psychological needs (competence satisfaction, autonomy satisfaction and relatedness satisfaction) and may increasesocial anxiety in children; (c) Distancing and protection will decrease the basic psychological needs (competence satisfaction, autonomy satisfaction and relatedness satisfaction) and will decrease the experience of social anxiety in children; (d) Credibility of perceived information will promote the existence of higher levels of social anxiety in children.

Materials and Methods

Participants

The study design is cross-sectional, and the sample collection was performed through non-probabilistic and incidental sampling. The sample consisted of 58 schoolchildren (M age = 10.18; SD = 0.77; 36 boys and 22 girls) from the Valencian Community in Spain. The children’s hours of study were 7.98 h per week, 10 suffered a COVID-19 infection (17.24%), 31 had a relative infected by COVID-19 (53.45%), 17 participants and their relatives were not infected by COVID-19 (29.31%), 41 practiced sports and physical activity (70.69%) and 17 were sedentary (29.31%). In particular, 18 were associated athletes (30.51%), 14 competed at regional level (24.14%) and five competed at national level (8.62%).

The sample was selected according to the study’s goals as it was intended to examine whether COVID-19 causal factors perceived by children predicted basic psychological needs and social anxiety; and if basic psychological needs predicted social anxiety. The only inclusion criterion was to evaluate schoolchildren as we wanted to examine the effects of the epidemic on the aforementioned variables in schoolchildren.

Instruments

To measure the perception of the population regarding the pandemic causality, it was utilized the “Questionnaire about the causality factors of COVID-19 pandemic” (CPFC-COVID-19) (Prieto, 2022). The questionnaire consists of 20 items with a Likert-type scale ranging from 1. totally disagree, 2. disagree, 3. agree and 4. totally agree. The CPFC-COVID-19 has four dimensions: 1) “Social distancing and protection” (DSP) with six items (e.g., I think that if I go down the street without a mask, I put no one at risk of contagion); 2) “Perceived psychological impact” (IPP) with five items (e.g., I think that stress influences the risk of contagion by COVID-19); 3) “Skepticism” (ES) with four items (e.g., I am sure I will not get infected if I go with my group of friends); and 4) “Credibility of the information received” (CIP) with five items (e.g., I think that the population is not well informed about the political and social situation related to curbing COVID). As Cronbach alpha increases with a high number of items, it was taken as an internal consistency marker the mean inter-item correlation (Clark & Watson, 1995; DeVellis, 2003). Several scholars have selected this measure in which an inter-item correlation higher than 0.15 is an adequate cut-off point. Thus, in this study the mean inter-item correlation of the distinct factors was: 0.20 (DSP), 0.27 (IPP), 0.50 (ES) and 0.40 (CIP). Finally, it is important to highlight that the authors have permission to use this instrument from the copyright holders and the instrument has shown appropriate reliability and validity in a previous study (Prieto, 2022).

The Spanish version of the Basic Needs Satisfaction in General Scale (BNSG-S) (Costa et al., 2022; Gagné, 2003). The original scale is made up of 21 items that measure the satisfaction of competency needs (r = 0.16; six items, e.g., “Most days I feel that I am successful in what I do”), autonomy satisfaction (r = 0.20; seven items, e.g., “I feel that I am free to decide for myself how to live my life”) and relatedness (r = 0.34; eight items, e.g., “I really like the people that I interact with”). In each of the dimensions, there were three items written in a negative way. The participants had to answer all the items thinking about how they related to their life and indicating how true they were for them on a Likert-type scale from 1 (not true) to 7 (totally true). Finally, it is important to highlight that the authors have permission to use this instrument from the copyright holders and the instrument has shown appropriate reliability and validity in previous studies (Gagné, 2003; González-Cutre et al., 2015).

The Psychological Needs Thwarting Scale was utilized (EFNP) (Bartholomew et al., 2011; Sicilia, Ferriz & Sáenz-Álvarez, 2013). The original scale is preceded by the phrase “In exercise…” but in this study to be adapted by life in general it was changed to: “In my life…”. It consists of 12 items that are distributed in four items for each of the three subscales in which it is composed: a) Frustration of the need for autonomy (r = 0.51; e.g., “ I feel obligated to follow the decisions of others”); b) Frustration of the need for competence (r = 0.48; e.g., “there are situations in which I feel incapable”); and c) Frustration of the need for relatedness (r = 0.25; e.g., “I feel that other people don’t like me”). For each item, the participants must indicate their response on a 7-point Likert scale ranging from 1 = do not agree at all, 2 = very slightly agree, 3 = slightly agree, 4 = moderately agree, 5 = agree, 6 = strongly agree to 7 = very strongly agree. Higher scores indicate a higher level of frustration of psychological needs. Finally, it is important to highlight that the authors have permission to use this instrument from the copyright holders and the instrument has shown appropriate reliability and validity in previous studies (Bartholomew et al., 2011; Sicilia, Ferriz & Sáenz-Álvarez, 2013).

The social anxiety questionnaire (CASO-N24) (Caballo et al., 2012) was used in this study. This measure presents two versions, one aimed at boys and the other at girls, both with 24 items with a Likert-type response scale from 1 to 4, being 1: not at all, 2: a little, 3: quite a lot, and 4: a lot. The items of this questionnaire were obtained from a review of the literature on childhood fears and problematic social situations in children who came for consultation for social phobia. The questionnaire contemplates six factors: Public speaking/Interaction with teachers (r = 0.53); Interaction with the opposite sex (r = 0.63); Being embarrassed or ridiculed (r = 0.37); Assertive expression of annoyance or anger (r = 0.49); Interaction with strangers (r = 0.54); and Acting in public (r = 0.33). Finally, it is important to highlight that the authors have permission to use this instrument from the copyright holders and it has shown appropriate reliability and validity in previous studies (Caballo et al., 2012).

Procedure

The study was approved by the ethics committee of Universidad Internacional de La Rioja (UNIR) whose approval code is PI024/2022. The study complied with the ethical standards of the American Psychology Association and with the rules of the Declaration of Helsinki (World Medical Association, 2013). Before beginning the process of administering the questionnaires, permission was requested from the school centre and the student’s parents. A signed in-person informed consent was obtained from all participants’ parents.

In the informed consent, the confidentiality and anonymity of the data was ensured. The questionnaires were administered individually through a Google forms link at the computer room in the school hours from February 2022 to April 2022. The administration of the questionnaires was carried out by teachers and research staff to assist with any possible doubts that might arise during the procedure. The length of the survey was around 20 min. Once the questionnaires were finished, the data were directly recorded in Google’s database.

Data analyses

The statistical software R (4.1.1) was utilized to conduct the distinct statistical analyses. To analyze the relationship across the pandemic causality factors, basic psychological needs satisfaction and frustration, and social anxiety, a Partial Least Square Path Modelling (PLS-PM) approach was taken (Sánchez, 2013). PLS-PM is a variance-based structural equation modelling technique which is not constrained by distributional assumptions (Martinent et al., 2019; Nicolas, Drapeau & Martinent, 2017; Sánchez, 2013). Thus, it signifies that PLS-PM is a good technique to work with little sample sizes. The significance of the parameter estimates is assessed by constructing 95% bias-corrected percentile confidence interval based on a bootstrap procedure with 100 replications (Martinent et al., 2019). Concerning all the variables (causality factors of COVID-19, basic psychological need satisfaction and needs frustration), two parcels were made (Coffman & MacCallum, 2005) in each factor. To create the parcels it was used a subset-item-parcel approach in which there were a random aggregate of items. This approach of parceling was taken because it improves the measurement properties of constructs, simplifies model complexity, enhances normality, reduces multicollinearity, and provides stable estimates in PLS-PM, especially when dealing with small sample sizes (Matsunaga, 2008).

In addition, different steps were followed to run the distinct analyses. First, the psychometric properties of the variables measured were examined. The indexes measured to ensure sufficient quality were: Standardized factor loadings, composite reliability values (ρ), average variance extracted (AVE) values and eigenvalue analysis of the correlation matrix of each set of manifest variables (Martinent et al., 2019; Nicolas, Drapeau & Martinent, 2017). Standardized factor loadings higher than 0.40, (Martinent et al., 2019) ρ values greater than 0.70, (Raykov, 2001) AVE values equal or greater than 0.50, (Fornell & Larcker, 1981) the first eigenvalue larger than 1 and the second one smaller than 1 (Nicolas, Drapeau & Martinent, 2017) indicate acceptable reliability of latent and manifest scores. Second, the structural model was tested to examine the connection among the latent variables examined in the study.

Results

Descriptive statistics and correlations among the variables

Table 1 shows the descriptive statistics and correlations among the variables. The results revealed low-medium scores in social distancing and protection (DSP), perceived psychological impact (IPP), Skepticism (ES), credibility (CIP), autonomy frustration (THAUT), competence frustration (THCOMP), relatedness frustration (THRELA), public speaking/interaction with teachers (HABPUB), interaction with the opposite sex (SEX), being embarrassed or ridiculed (EVI), assertive expression of annoyance or anger (ASERT), interaction with strangers (DESCON) and acting in public (ACT). On the other hand, high scores were found in autonomy satisfaction, competence satisfaction and relatedness satisfaction. Moreover, correlational analyses showed that there was no multicollinearity between the study variables as correlations ranged from −0.54 to 0.71 and none of the confidence intervals (i.e., r ± two standard errors) were close to 1.0.

Table 1 Descriptive statistics and correlations among the variables.

	1	2	3	4	5	6	7	8	9	10	11	12	13	14	15	16	
1. DSP	–																
2. IPP	0.227	–															
3 ES	0.476**	0.523**	–														
4. CIP	0.504**	0.099	0.168	–													
5. AUT	−0.257	−0.093	−0.016	−0.360**	–												
6. COM	−0.546**	−0.270*	−0.213	−0.493**	0.532**	–											
7. RELA	−0.356**	−0.383**	−0.161	−0.383**	0.618**	0.717**	–										
8. THAUT	0.318*	0.413**	0.240	0.350**	−0.440**	−0.570**	−0.613**	–									
9. THCOMP	0.278*	0.343**	0.301*	0.347**	−0.494**	−0.455**	−0.598**	0.685**	–								
10. THRELA	0.245	0.432**	0.216	0.305*	−0.348**	−0.474**	−0.647**	0.695**	0.689**	–							
11. HABPUB	0.162	0.006	−0.077	0.109	−0.081	−0.315*	−0.174	0.372**	0.087	0.295*	–						
12. SEX	−0.190	−0.256	0.059	0.073	0.226	0.173	0.080	0.059	0.018	−0.128	0.077	–					
13. EVI	0.000	0.027	0.095	0.166	−0.048	−0.092	−0.218	0.385**	0.354**	0.219	0.223	0.580**	–				
14. ASERT	0.080	0.016	−0.004	0.036	−0.098	−0.347**	−0.199	0.376**	0.266*	0.312*	0.640**	0.170	0.483**	–			
15. DESCON	0.023	−0.084	0.047	0.329*	−0.092	−0.136	−0.110	0.416**	0.323*	0.311*	0.424**	0.397**	0.476**	0.401**	–		
16. ACT	0.037−	0.098	0.093	0.277*	0.054	−0.053	0.005	0.163	0.179	0.087	0.293*	0.597**	0.426**	0.285*	0.575**	–	
Mean	1.85	1.94	2.21	2.16	4.83	5.02	5.45	2.11	2.19	2.26	2.41	2.67	2.25	2.26	2.28	2.43	
Standard Deviation	0.57	0.61	0.57	0.46	1.01	0.99	1.17	0.89	0.89	0.86	0.92	1.04	0.80	0.87	0.81	0.84	
Skewness	0.58	0.67	0.55	−0.16	0.05	0.12	−0.30	1.04	0.98	0.64	0.01	−0.16	0.29	0.33	0.16	0.21	
Kurtosis	−0.26	1.05	0.72	0.22	−1.14	−0.69	−1.29	1.35	0.84	0.50	−0.97	−1.41	−0.58	−0.72	−0.76	−1.01	
Notes:

* p < 0.05.

** p < 0.01.

DSP, Social distancing and protection; IPP, Perceived psychological impact; ES, Skepticism; CIP, Credibility; AUT, Autonomy satisfaction; COMP, Competence satisfaction; RELA, Relatedness satisfaction; THAUT, Autonomy frustration; THCOMP, Competence frustration; THRELA, Relatedness frustration; HABPUB, Public speaking/Interaction with teachers; SEX, Interaction with the opposite sex; EVI, Being embarrassed or ridiculed; ASERT, Assertive expression of annoyance or anger; DESCON, Interaction with strangers; ACT, Acting in public.

Inner PLS-PM model

Table 2 shows the results of the inner PLS-PM model in which the distinct indexes provided enough markers of the reliability and validity of the model. In particular, the standardized factor loadings ranged between 0.60 and 0.96 (M = 0.86; SD = 0.11), the ρ values ranged between 0.72 and 0.95 (M = 0.86; SD = 0.07), the AVE values ranged from 0.60 to 0.91 (M = 0.74; SD = 0.15) whereas the first eigenvalues ranged from 1.21 to 1.78 (M = 1.55; SD = 0.18) and the second eigenvalues ranged from 0.16 to 0.83 (M = 0.48; SD = 0.20).

Table 2 Psychometric properties of the study variables.

Variables	Construct level statistics	Items/Parcels	SFL	
1. DSP	ʎ1 = 1.39; ʎ2 = 0.61	1	0.90	
ρ = 0.82; AVE = 0.70	2	0.73	
2. IPP	ʎ1 = 1.49; ʎ2 = 0.51	1	0.89	
ρ = 0.85; AVE = 0.74	2	0.82	
3 ES	ʎ1 = 1.21; ʎ2 = 0.79	1	0.60	
ρ = 0.75; AVE = 0.60	2	0.92	
4. CIP	ʎ1 = 1.39; ʎ2 = 0.83	1	0.80	
ρ = 0.72; AVE = 0.60	2	0.60	
5. AUT	ʎ1 = 1.68; ʎ2 = 0.31	1	0.88	
ρ = 0.91; AVE = 0.83	2	0.94	
6. COM	ʎ1 = 1.43; ʎ2 = 0.57	1	0.88	
ρ = 0.83; AVE = 0.71	2	0.80	
7. RELA	ʎ1 = 1.84; ʎ2 = 0.16	1	0.95	
ρ = 0.95; AVE = 0.91	2	0.96	
8. THAUT	ʎ1 = 1.76; ʎ2 = 0.23	1	0.93	
ρ = 0.93; AVE = 0.88	2	0.94	
9. THCOMP	ʎ1 = 1.62; ʎ2 = 0.38	1	0.91	
ρ = 0.89; AVE = 0.80	2	0.88	
10. THRELA	ʎ1 = 1.23; ʎ2 = 0.76	1	0.85	
ρ = 0.76; AVE = 0.61	2	0.70	
11. HABPUB	ʎ1 = 1.68; ʎ2 = 0.32	1	0.95	
ρ = 0.91; AVE = 0.83	2	0.86	
12. SEX	ʎ1 = 1.78; ʎ2 = 0.22	1	0.95	
ρ = 0.94; AVE = 0.88	2	0.93	
13. EVI	ʎ1 = 1.55; ʎ2 = 0.44	1	0.94	
ρ = 0.87; AVE = 0.76	2	0.80	
14. ASERT	ʎ1 = 1.63; ʎ2 = 0.37	1	0.91	
ρ = 0.89; AVE = 0.81	2	0.88	
15. DESCON	ʎ1 = 1.67; ʎ2 = 0.33	1	0.93	
ρ = 0.90; AVE = 0.83	2	0.89	
16. ACT	ʎ1 = 1.47; ʎ2 = 0.52	1	0.75	
ρ = 0.84; AVE = 0.72	2	0.93	
Note:

ʎ1: ith eigenvalue of the item correlation matrix; ρ: composite reliability; AVE: average variance extracted; SFL: standardized factor loadings. All SFLs were significant at p < 0.001; DSP = Social distancing and protection; IPP = Perceived psychological impact; ES = Skepticism; CIP = Credibility; AUT = Autonomy satisfaction; COMP = Competence satisfaction; RELA = Relatedness satisfaction; THAUT = Autonomy frustration; THCOMP = Competence frustration; THRELA = Relatedness frustration; HABPUB = Public speaking/Interaction with teachers; SEX = Interaction with the opposite sex; EVI = Being embarrassed or ridiculed; ASERT = Assertive expression of annoyance or anger; DESCON = Interaction with strangers; ACT = Acting in public.

PLS-PM model (relationships between the latent variables)

Table 3 shows the results of the PLS-PM model (Fig. 1: i.e., relationships between the latent variables). The results revealed that social distancing and protection significantly negatively predicted competence satisfaction (β = −0.13; p < 0.05). Besides, perceived psychological impact significantly negatively predicted relatedness satisfaction (β = −0.35; p < 0.05), significantly positively autonomy frustration (β = 0.34; p < 0.05) and competence frustration (β = 0.26; p < 0.05). In addition, perceived psychological impact significantly negatively predicted interaction with the opposite sex (β = −0.36; p < 0.05). Moreover, autonomy satisfaction significantly positively predicted interaction with the opposite sex (β = 0.48; p < 0.05). On the other hand, autonomy frustration significantly positively predicted public speaking/interaction with teachers (β = 0.44; p < 0.05), significantly positively interaction with the opposite sex (β = 0.35; p < 0.05), significantly positively being embarrassed or ridiculed (β = 0.46; p < 0.05) and significantly positively interaction with strangers (β = 0.49; p < 0.05).

Table 3 Structural model.

Variables	Total sample (n = 58)	
BME	CI	
DSP -> AUT	−0.13	[−0.45 to 0.22]	
DSP -> COM	−0.40*	[−0.70 to −0.01]	
DSP -> RELA	−0.19	[−0.48 to 0.07]	
DSP -> THAUT	0.11	[−0.18 to 0.42]	
DSP -> THCOMP	0.12	[−0.14 to 0.42]	
DSP -> THRELA	0.08	[−0.19 to 0.37]	
DSP -> HABPUB	0.09	[−0.26 to 0.52]	
DSP -> SEX	−0.33	[−0.69 to 0.05]	
DSP -> EVI	−0.21	[−0.50 to 0.11]	
DSP -> ASERT	−0.13	[−0.47 to 0.35]	
DSP -> DESCON	−0.11	[−0.42 to 0.21]	
DSP -> ACTPUBLI	−0.18	[−0.50 to 0.17]	
IPP -> AUT	−0.14	[−0.47 to 0.07]	
IPP -> COM	−0.14	[−0.39 to 0.16]	
IPP -> REL	−0.35*	[−0.66 to −0.09]	
IPP -> THAUT	0.34*	[0.08–0.57]	
IPP -> THCOMP	0.26*	[0.00–0.48]	
IPP -> THRELA	0.37	[−0.02 to 0.69]	
IPP -> HABPUB	−0.13	[−0.38 to 0.17]	
IPP -> SEX	−0.36*	[−0.69 to −0.08]	
IPP -> EVI	−0.16	[−0.51 to 0.16]	
IPP -> ASERT	−0.16	[−0.49 to 0.28]	
IPP -> DESCON	−0.25	[−0.61 to 0.12]	
IPP -> ACTPUBLI	−0.22	[−0.61 to 0.24]	
ES -> AUT	0.10	[−0.30 to 0.46]	
ES -> COM	0.04	[−0.37 to 0.46]	
ES -> RELA	0.13	[−0.25 to 0.57]	
ES -> THAUT	−0.03	[−0.42 to 0.30]	
ES -> THCOMP	0.06	[−0.38 to 0.42]	
ES -> THRELA	−0.04	[−0.38 to 0.31]	
ES -> HABPUB	−0.10	[−0.43 to 0.26]	
ES -> SEX	0.30	[−0.22 to 0.64]	
ES -> EVI	0.11	[−0.21 to 0.45]	
ES -> ASERT	−0.01	[−0.39 to 0.34]	
ES -> DESCON	0.04	[−0.29 to 0.32]	
ES -> ACTPUBLI	0.13	[−0.30 to 0.58]	
CIP -> AUT	−0.20	[−0.60 to 0.48]	
CIP -> COM	−0.10	[−0.47 to 0.44]	
CIP -> RELA	−0.17	[−0.50 to 0.40]	
CIP -> THAUT	0.15	[−0.41 to 0.50]	
CIP -> THCOMP	0.18	[−0.33 to 0.50]	
CIP -> THRELA	0.18	[−0.32 to 0.46]	
CIP -> HABPUB	−0.02	[−0.31 to 0.26]	
CIP -> SEX	0.28	[−0.34 to 0.68]	
CIP -> EVI	0.14	[−0.29 to 0.50]	
CIP -> ASERT	0.06	[−0.28 to 0.42]	
CIP -> DESCON	0.20	[−0.39 to 0.52]	
CIP -> ACTPUBLI	0.27	[−0.42 to 0.71]	
AUT -> HABPUB	0.05	[−0.33 to 0.42]	
AUT -> SEX	0.48*	[0.12–0.92]	
AUT -> EVI	0.30+	[−0.001 to 0.60]	
AUT -> ASERT	0.07	[−0.24 to 0.42]	
AUT -> DESCON	0.16	[−0.15 to 0.58]	
AUT -> ACTPUBLI	0.23	[−0.28 to 0.61]	
COM -> HABPUB	−0.24	[−0.61 to 0.09]	
COM -> SEX	0.20	[−0.15 to 0.53]	
COM -> EVI	0.13	[−0.32 to 0.51]	
COM -> ASERT	−0.45	[−0.83 to −0.04]	
COM -> DESCON	0.03	[−0.30 to 0.41]	
COM -> ACTPUBLI	−0.08	[−0.47 to 0.34]	
REL -> HABPUB	0.13	[−0.26 to 0.53]	
REL -> SEX	−0.40+	[−0.92 to 0.007]	
REL -> EVI	−0.31	[−0.76 to 0.37]	
REL -> ASERT	0.31	[−0.17 to 0.93]	
REL -> DESCON	0.17	[−0.25 to 0.60]	
REL -> ACTPUBLI	0.04	[−0.49 to 0.48]	
THAUT -> HABPUB	0.44*	[0.11–0.67]	
THAUT -> SEX	0.38*	[0.11–0.67]	
THAUT -> EVI	0.46*	[0.13–0.78]	
THAUT -> ASERT	0.26	[−0.04 to 0.58]	
THAUT -> DESCON	0.49*	[0.19–0.94]	
THAUT -> ACTPUBLI	0.16	[−0.21 to 0.47]	
THCOMP -> HABPUB	−0.35	[−0.76 to 0.02]	
THCOMP -> SEX	0.05	[−0.29 to 0.60]	
THCOMP -> EVI	0.23	[−0.12 to 0.78]	
THCOMP -> ASERT	0.04	[−0.33 to 0.51]	
THCOMP -> DESCON	0.14	[−0.22 to 0.66]	
THCOMP -> ACTPUBLI	0.11	[−0.42 to 0.58]	
THRELA -> HABPUB	0.28	[−0.17 to 0.70]	
THRELA -> SEX	−0.30	[−0.63 to 0.11]	
THRELA -> EVI	−0.17	[−0.56 to 0.16]	
THRELA -> ASERT	0.16	[−0.24 to 0.59]	
THRELA -> DESCON	0.15	[−0.29 to 0.65]	
THRELA -> ACTPUBLI	−0.03	[−0.45 to 0.38]	
Notes:

* p<0.05.

+ p ≤ 0.09.

DSP = Social distancing and protection; IPP = Perceived psychological impact; ES = Skepticism; CIP = Credibility; AUT = Autonomy satisfaction; COMP = Competence satisfaction; RELA = Relatedness satisfaction; THAUT = Autonomy frustration; THCOMP = Competence frustration; THRELA = Relatedness frustration; HABPUB = Public speaking/Interaction with teachers; SEX = Interaction with the opposite sex; EVI = Being embarrassed or ridiculed; ASERT = Assertive expression of annoyance or anger; DESCON = Interaction with strangers; ACT = Acting in public.

Figure 1 Relationships among the studied variables.

Discussion

The goals of the present study were to examine whether COVID-19 causal factors perceived by children predicted basic psychological needs and social anxiety; and if basic psychological needs predicted social anxiety.

Social distancing, protection and competence

Results revealed that social distancing and protection significantly negatively predicted competence satisfaction. These results are in line with the study of Scheid et al. (2020) as it states that the mask use and distance measures may influence the perception of competence satisfaction in schoolchildren. These results make sense because compliance with mask use and distance measures may hinder how children work at school and the way in which they interact with the environment and their peers (Coelho et al., 2022). For instance, interacting with children when working on a group task is different when wearing a mask or not. Thus, the experience of working at school and respecting the rules may hinder competence satisfaction as there are several measures to be respected that may worsen competence.

Perceived psychological impact and basic psychological needs

Perceived psychological impact significantly negatively predicted relatedness satisfaction, significantly positively autonomy frustration, and competence frustration. This is in line with previous studies in which the distinct outcomes caused by the epidemic have triggered a different influence on basic psychological needs (Šakan, Žuljević & Rokvić, 2020).

In this sense, perceived psychological impact can negatively influence a person’s sense of autonomy, making them feel less able to control their own actions and decisions. For example, people who experience high levels of perceived psychological impact may feel compelled to avoid social situations due to fear of judgment or negative evaluation. This avoidance can be seen as a lack of control over their own lives, which can diminish their sense of autonomy. In another way, perceived psychological impact can undermine the sense of competence, especially in social contexts, where the perception of not meeting social expectations can be debilitating. For instance, studies have shown that a lack of perceived competence in social interactions is closely related to high levels of social anxiety (Weeks et al., 2008). These results are in line with a previous study that highlights the increased difficulties to interact with others in pandemic times due to the impact on fear and anxiety (Calbi et al., 2021).

Perceived psychological impact and social interaction

In addition, results revealed that perceived psychological impact significantly negatively predicted interaction with the opposite sex. Specifically, may lead to avoiding social interactions to prevent situations that might exacerbate negative feelings, due to the risk of contagion. In this sense, the increase in stress, anxiety and depressive symptoms related to contagion (Chen et al., 2020) could generate, due to preventive measures, social conflict, isolation, devaluation, rejection and exclusion (Slavich, 2020). This avoidant behavior is a common characteristic of social anxiety, which has been shown to negatively affect interpersonal relationships (Alden & Taylor, 2004).

Otherwise, the interaction with the other sex may help to reduce those perceived psychological impact experienced in the epidemics, disconnecting and helping to reduce the maladaptive outcomes of the causal factors provoked by the pandemic (Luo et al., 2022). In this sense, appropriate social supports with alternative means for interpersonal communication (e.g., social media), frequent contact with colleagues, can alleviate stress (Yan et al., 2021) and promote interaction between people. Thus, despite the increase of fear and anxiety which minimize the interaction with others, it is needed to emphasize in children the need to socialize with others respecting the rules as a way to enhance their wellbeing.

Social interaction, autonomy and social anxiety

According to self-determination theory, higher autonomy satisfaction would be expected to lead to better social outcomes, and conversely, autonomy frustration would be associated with negative outcomes such as social avoidance. However, our findings indicate that social interaction is not significantly affected by autonomy levels, which challenges the typical expectations of self-determination theory. One possible theoretical explanation is that both autonomy satisfaction and frustration may motivate social interactions, albeit for different reasons. Autonomy satisfaction might drive positive relationships due to feelings of competence and security, whereas autonomy frustration might generate interactions in an attempt to compensate for a lack of social control or support. Therefore, both can be present in all relational contexts, as suggested by the cited study (Sibley & Overall, 2008).

Moreover, autonomy satisfaction and frustration significantly positively predicted social anxiety in the interaction with the opposite sex. This finding suggests that social anxiety in the interaction with the opposite sex may depend on other factors different than autonomy satisfaction or frustration. It is possible that social interactions with the opposite sex may be maintained regardless of autonomy level, due to external motivations such as social norms, interpersonal attraction, or the need for affiliation, such as the frequency of feeling supported by using technologies, seeing friends, creating challenges or carrying out the same activities simultaneously (Muñoz-Fernández & Rodríguez-Meirinhos, 2021). Furthermore, results revealed that autonomy frustration was positively related to anxiety to being embarrassed or ridiculed. The pandemic context likely influenced results, as increase stress (Chen et al., 2020), social isolation, and uncertainty may have heightened social conflict, rejection and exclusion (Slavich, 2020), regardless of frustration and autonomy satisfaction, with women being more susceptible to COVID-19-related mental health problems (Yalçın et al., 2021), and gender differences being evident in exposure to potentially traumatic events (Street & Dardis, 2018). Besides, autonomy frustration positively predicted anxiety in public speaking/interaction with teachers and significantly positively interaction with strangers. In particular, when a person experiences frustration in their autonomy, they may feel they do not have control over their decisions. As a consequence, they may develop social anxiety, especially if they struggle to maintain social relationships, in line with the following study (Gao et al., 2022). On the other hand, those who are restricted in their social interaction, for example, due to pandemic isolation, may experience fear and social anxiety due to reduced exposure to frequent social situations (Rahm-Knigge, Prince & Conner, 2021). In this sense, encouraging autonomy and socialization, especially in educational contexts and during periods of isolation such as a pandemic, is crucial for promoting emotional well-being and prevent disorders such as social anxiety.

Some of the limitations are that the usage of self-report measures may lead to some bias such as: social desirability, acquiescence and dishonestly (Latkin et al., 2017). Nevertheless, the measures taken were the most appropriate to the population selected. On the other hand, the utilization of a specific school may hinder the generalization of the results to other different populations and contexts. As such, the results should be understood in the context in which the study was conducted. Finally, the small sample size analyzed in this study hinders the generalization of the findings obtained and minimizes the power of the conclusions shown. Thus, in future studies, it would be interesting to increase the sample size, adding more schools from different regions and students with diverse sociodemographic characteristics to facilitate the generalization of the findings obtained. Nevertheless, the results obtained are valuable as they show a big picture of the relationship among COVID-19 causal factors, basic psychological needs and social anxiety, which may help in future epidemics to make decisions to enhance children’s well-being. It is also worth noting that the literature also indicates that female adolescents show higher levels of social anxiety than male adolescents, so future research could make these comparisons (Prieto, 2020).

Conclusions

Perceived psychological impact was the causal factor that revealed a higher impact on basic psychological needs. In addition, autonomy frustration was the variable that revealed a higher impact on social anxiety variables. These results shed light on how the causal factors of COVID-19 in children and their implication in other well-being variables (basic psychological needs and social anxiety). The main conclusions of the study are evident:

- Social distancing and protection negatively predicts competence satisfaction, implying that measures like mask-wearing and social distancing affect how children perceive their abilities and interactions in their environment. For example, how adherence to distancing rules might affect classroom dynamics or group projects can be a possible scenario where social distancing and protection measures directly impact children’s activities or interactions.

- Perceived psychological impact affects each of these variables separately: relatedness satisfaction: perceived psychological impact might strain interpersonal relationships among children, perhaps due to increased stress or changes in social dynamics which prevents the satisfaction of relatedness; autonomy frustration: children’s sense of autonomy frustration is strengthened during the pandemic, such as restrictions on movement or decision-making; and competence frustration: challenges imposed by the pandemic (like remote learning) hinder children’s sense of competence satisfaction.

- Specific suggestions or recommendations: educators could focus on clear communication strategies to mitigate anxiety among students; psychologists might prioritize interventions that enhance children’s sense of autonomy satisfaction, competence satisfaction and relatedness satisfaction despite pandemic challenges; and Health professionals could collaborate with schools to ensure accurate and supportive messaging reaches children and their families.

- These recommendations could be implemented in real-world settings, considering practical constraints and opportunities. There is a potential paradigm shift in understanding pandemic impacts on children. This study redefines our understanding of how pandemics, like COVID-19, impact children’s psychological well-being. It highlights that measures such as social distancing and protective protocols negatively affect children’s perceived competence, limiting their social and academic interactions. Perceived Psychological Impact emerges as pivotal, influencing basic psychological needs and social anxiety. Perceived psychological impact negatively impacts peer relationship satisfaction but positively correlates with autonomy frustration and competence frustration, revealing how individual stress perceptions shape children’s interpersonal dynamics and self-image during pandemics. Autonomy satisfaction is identified as crucial for positive outcomes, predicting increased social engagement and marginal emotional well-being.

- In terms of implications, the study urges tailored interventions recognizing children’s diverse responses to pandemics. Specifically, for education professionals, it may be convenient to carry out actions in an interdisciplinary way focused on developing critical thinking, minimizing skepticism in pandemic situations and others that require it. Besides, it is considered pertinent to promote the development and acquisition of basic psychological needs, both from each subject and in coordination with the guidance department. More related to physical education, physical activity programs should be established to develop resilience and stimulate communication and social interactions among peers. It would also be advisable to hold informative talks for the family, trying to raise awareness about the importance of meeting the basic psychological needs of children and about the risks derived from social anxiety. Likewise, for psychologists and other health professionals, a measure can be to create and optimize training, communication and counseling channels to meet basic psychological needs and minimize the perceived psychological impact as well as social anxiety. It would also be interesting to reinforce self-concept, self-confidence and self-esteem, thus favoring interaction with classmates and teachers, speaking in public and minimizing the fact of feeling embarrassed or ridiculed. On the other hand it is advisable to develop initiatives such as mindfulness or yoga that help develop self-control and regulate social anxiety. All these measures will be more effective if cooperation and action are achieved between the different agents involved. Ultimately, the study represents a paradigm shift by underscoring the critical role of mental health and socio-emotional development in pandemic preparedness and response. This shift guides policies and interventions that prioritize children’s holistic development and mental well-being during health crises.

Supplemental Information

Supplemental Information 1 Raw Data.

Supplemental Information 2 STROBE checklist.

Supplemental Information 3 Codebook.

Supplemental Information 4 Coding line.

Additional Information and Declarations

Competing Interests

The authors declare that they have no competing interests.

Author Contributions

Higinio González-García conceived and designed the experiments, performed the experiments, analyzed the data, prepared figures and/or tables, authored or reviewed drafts of the article, and approved the final draft.

Leandro Álvarez-Kurogi conceived and designed the experiments, performed the experiments, analyzed the data, prepared figures and/or tables, authored or reviewed drafts of the article, and approved the final draft.

Joel Prieto Andreu conceived and designed the experiments, performed the experiments, analyzed the data, prepared figures and/or tables, authored or reviewed drafts of the article, and approved the final draft.

Javier Tierno Cordón conceived and designed the experiments, performed the experiments, analyzed the data, prepared figures and/or tables, authored or reviewed drafts of the article, and approved the final draft.

Rosario Castro López conceived and designed the experiments, performed the experiments, analyzed the data, prepared figures and/or tables, authored or reviewed drafts of the article, and approved the final draft.

Jesús Salas Sánchez conceived and designed the experiments, performed the experiments, analyzed the data, prepared figures and/or tables, authored or reviewed drafts of the article, and approved the final draft.

Human Ethics

The following information was supplied relating to ethical approvals (i.e., approving body and any reference numbers):

Universidad Internacional de La Rioja (PI024/2022).

Ethics

The following information was supplied relating to ethical approvals (i.e., approving body and any reference numbers):

Universidad Internacional de La Rioja (UNIR).

Data Availability

The following information was supplied regarding data availability:

The raw data is available in the Supplemental File.

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
