# Peer review of "Relationships among COVID-19 causal factors perceived by children, basic psychological needs and social anxiety"

_PeerJ, doi:10.7717/peerj.18828_

## Round 0.1 · original submission · Major Revisions

The manuscript requires improvement in terms of writing quality and clarification on specific issues, pointed out by the reviewers. Please, be careful and address reviewers’ doubts point by point

·

Basic reporting

Dear researchers, hello
Your manuscript addresses an interesting topic and is a valuable research. Congratulations on this. However, the manuscript requires significant improvement in terms of writing quality as well as native editing. Please consider the following points to enhance the overall quality of your manuscript.

Abstract
- In line 26, the word "relatedness" is correct, not "relationship".
- In line 27, the phrases “autonomy frustration” and “competence frustration” are correct. Please correct throughout the manuscript.
- There is no need to report results with marginal significant in the abstract.

Introduction
- From lines 42 to 51, you have explained the COVID-19 causal factors, but you have not provided any reference. Please add the reference.
- From line 56 onwards, first explain the self-determination theory and basic psychological needs. Then point out the consequences of satisfaction and frustration of basic psychological needs. Next, bring what you wrote about the effect of the causal factors of the epidemic on basic psychological needs.
- In lines 61 to 63, the sentences are unintelligible and need rewriting.
- From line 76 onwards, first define social anxiety and then bring what you wrote about the effect of the causal factors of the epidemic on social anxiety. In other words, move the content written in lines 83 to 88 to here.
- Define each factor mentioned in line 90 briefly. Also write the abbreviation of the phrase in front of it.
- State the purpose of the study only once at the end of the introduction.
- Specify which theoretical framework is referred to in line 101.
- Rewrite hypotheses from line 102 onwards. By autonomy, competence, and relatedness, do you mean the satisfaction of these three needs or frustration? If you mean satisfaction of needs, why do you assume that DSP and ES increase them? (Please specify throughout the text whether you mean satisfaction of these needs or frustration).

References
-Please use EndNote. In some references, the year of the study is written incorrectly.

Experimental design

Method
This section of the manuscript is poorly written. Please answer the following questions in the manuscript.
- How was the sample size determined in this study? How did you ensure that 58 participants were sufficient for this study?
- What sampling method was used?
- What were the inclusion and exclusion criteria for the research?
- In line 112, it is stated that participants studied an average of 7.98 hours per week. What is it written for?
- In line 112, it is mentioned that 10 individuals had a COVID-19 infection, and 31 had a relative infected by COVID-19. So what about the remaining 17 participants? Was this accounted for during data analysis? (Please report both numbers and percentages).
- In line 113, what is meant by "any sports modality"? Please provide accurate information on how many individuals engaged in physical activity and how many did not. Was this considered during data analysis?
- In the section introducing instruments, please first write and bold the title of the questionnaire. Then introduce the questionnaire by mentioning who designed it and how its validity and reliability are. Also report the results of Cronbach's alpha (you can include it in Table 1).
- Explain more about the data collection method.
- In line 177, you have repeated a sentence twice.
- The sentence in line 183 should be rewritten.

Validity of the findings

Results
- In line 198, it is preferable not to use only abbreviations of phrases. Write out the full phrase and provide the abbreviation in parentheses next to it.
- In line 199, instead of autonomy, competence and relatedness, write autonomy satisfaction, competence satisfaction and relatedness satisfaction. Throughout the text and tables, it should be corrected like this.
- In line 215, it is written that relatedness satisfaction positively predicts SEX, however, Table 3 shows a negative prediction.
- In this study, it was found that autonomy frustration positively predicts EVI. Bring this finding in the abstract section.
- In Table 3, the total sample (n=317) is written. What does this number refer to?

Discussion and conclusion
- This section of your manuscript requires a complete rewrite. Start by stating the purpose of the study and the findings obtained. Then, present studies that align with the findings of this study to demonstrate to the reader that your findings are well-supported. If there are studies that contradict your findings, report them as well and explain the reason for this inconsistency. You have not made any reference to studies that are in line or contradict with this study in your manuscript.
- Furthermore, you discussed only some of your findings. It is essential to thoroughly explain all findings related to the relationships between Covid-19 causal factors and basic psychological needs, Covid-19 causal factors and social anxiety, as well as basic psychological needs and social anxiety.
- In line 234, there appears to be a contradiction between your study's findings and your explanation.
- In line 239, concentrate on elucidating the findings that were statistically significant rather than marginally significant results.
- Please mention the small sample size of this study in research limitations. Additionally, include suggestions for future research in the form of research proposals.

·

Basic reporting

Abstract:
“Participants: 58 schoolchildren (Mage = 10.18; SD = 0.77; 36 boys, 22 girls).” - should be in results, not objectives,
Add background to abstract

Manuscript:
“ There are distinct authors that have examined those COVID-19 causal factors perceived by the population” in line 40-41 does not make sense please clarify
Describe the meaning and difference of “causal factor” and “perceived factor”
It gets very confusing as authors start talking about those as two different things and then start talking about “causal perceived factors”

The statement about the study's goals ("relationship between the pandemic causality in distinct variables related with wellbeing in schoolchildren") is somewhat vague. It would help to specify the hypotheses or research questions driving the study.
The sample size and demographic details are clearly stated.

Experimental design

It would be beneficial to include a brief rationale for why this particular sample size was chosen and any power analysis conducted to justify it
Each instrument used is described in detail, including the number of items, response scale, and dimensions measured. This is very thorough.
The reliability measures (e.g., mean inter-item correlation, Cronbach's alpha) are well-reported, but the text would benefit from an explicit statement on the overall reliability and validity of each instrument.
The description of the psychometric properties' assessment and the criteria for acceptable reliability (e.g., factor loadings, composite reliability, AVE) are thorough. However, a brief explanation of what each measure indicates and why it's important would enhance understanding for readers less familiar with these concepts.
The bootstrap procedure is mentioned, but the rationale for using 100 replications should be provided, as this number is relatively low.

Validity of the findings

The interpretation of how social distancing and protection (DSP) negatively predicted competence is logical, but the explanation could be expanded to include more specific examples or references to existing literature.
The discussion of how perceived psychological impact (IPP) affects various psychological needs and social anxiety could be more detailed. Each point (e.g., IPP's effect on relationships, autonomy thwarting, and competence thwarting) should be clearly separated and explained with examples or theoretical support.
The practical implications are well-highlighted, particularly regarding the importance of information received during epidemics. However, the discussion could be expanded to include specific recommendations for educators, psychologists, and health professionals.
The mention of a potential paradigm shift is intriguing but needs more elaboration. How does this study contribute to a new understanding of pandemic impacts on children? More detail would help clarify this point.

Reviewer 3 ·

Basic reporting

This study investigates how children's perceptions of COVID-19 causal factors impact their basic psychological needs and social anxiety. The findings indicate that the perceived psychological impact had the most significant negative effect on basic psychological needs, while autonomy thwarting had the greatest effect on social anxiety. The study provides valuable insights into the psychological effects of COVID-19 on children. However, addressing the following suggestions could enhance the quality and robustness of the research.

Background:
1. Line 45: Please use a consistent abbreviation throughout the paper. You have interchangeably used IPP and PPI. Choose one and use it consistently.

Materials & Methods:
1. Sample Recruitment: Describe the setting for sample recruitment, indicating whether a convenience sample or probability sampling approach was used and specifying the geographic region/specific school targeted.
In addition, the details about the number of children, mean age, sex, and history of infection should be provided at the beginning of the results section rather than in the methods section. Suggest presenting such information as Table 1.

Experimental design

1. Instruments: Provide the exact scales used in the instruments in the manuscript or supplemental materials. The description of the instruments is not sufficiently detailed for readers to understand the scale wording between 1 (not true) and 7 (totally true) (line 141).

2. PLS-PM Clarification: Spell out PLS-PM at its first mention (line 178). Additionally, specify the exact R packages and their versions used for PLS-PM to ensure reproducibility.

3. Parceling: The method mentions creating two parcels using random aggregates of items (line 184). While parceling can simplify the model, it may obscure relationships between individual items and constructs. Better explain and justify the rationale and method of aggregation.

4. Bootstrap: The current use of only 100 bootstrap replications is insufficient (line 183). Increase the number of bootstrap replications to at least 1000 to provide more stable and reliable confidence intervals.

5. Factor Loadings: The method states that standardized factor loadings higher than .40 are acceptable (line 190). However, loadings closer to .70 or higher are ideal for strong convergent validity. Justify your choice and include a sensitivity analysis with a more stringent threshold.

6. Eigenvalue Criterion: The criterion of the first eigenvalue larger than 1 and the second one smaller than 1 (line 191) is based on the Kaiser criterion, which has been criticized for potentially retaining too many factors. A more accurate approach, such as parallel analysis, could provide a more precise determination of the number of factors to retain.

7. Robustness Checks: Include additional robustness checks or sensitivity analyses to strengthen the credibility of the findings, especially given the small sample size and the use of parceling.

Validity of the findings

Discussion Section: Emphasize the limitation of the small sample size. The high number of statistical tests conducted with a very small sample size may not guarantee the power of the conclusions drawn from this study.

---

## Round 0.2 · Major Revisions

The authors addressed most of the reviewers’ comments but some issues should be clarified/improved. In summary, the reviewer found that the paper has inaccuracies in using terms related to satisfaction and frustration of psychological needs; ii) certain sentences and word choices need clarification for better understanding and consistency; iii) the relationship between autonomy satisfaction/frustration and social interaction needs further explanation; iv) several lines would benefit from improved phrasing for precision.

Furthermore, as far as I understood, the manuscript claims both autonomy satisfaction and autonomy frustration positively predict interaction with the opposite sex, suggesting that social interaction is not significantly impacted by autonomy levels. This contradicts typical self-determination theory expectations, where higher satisfaction leads to positive outcomes and higher frustration leads to negative ones. The authors need to provide a more in-depth explanation or additional theoretical grounding for these findings

Also, some sections could benefit from more structured presentation and clarity. For example, the discussion could be organized to differentiate between psychological needs satisfaction and frustration findings clearly.

The implications for educators, psychologists, and health professionals should be expanded, providing clear recommendations based on the study’s results

·

Basic reporting

- Please correct the sentence in line 28. In Table 3, it can be seen that social distancing and protection negatively predicts competence satisfaction, not competence frustration.
- In line 32, it seems that you should write autonomy satisfaction. Because it is written in line 34 that autonomy frustration positively predicts interaction with the opposite sex.
- The finding of your study was that social distancing and protection negatively predicts competence satisfaction, which seems reasonable. However, why did you write in line 120 that you assume social distancing will strengthen satisfaction of needs?
- In the previous hypothesis in line 118, it does not seem very logical that skepticism will increase both satisfaction of needs and social anxiety. You wrote in the text that frustration of basic psychological needs is associated with increased social anxiety. Please check the written hypotheses again.
- In line 261, instead of the following manuscript, it is better to write study of Scheid et al. Also, in line 301, instead of the previous article, it is better to write the previous study.
- In line 278, doesn't the fact that IPP negatively predicted SEX mean that with increasing IPP, anxiety when interacting with the opposite sex decreases? If it means that, there is a little contradiction in your explanation.
- The material written in lines 282 to 290 seems to be more related to the relationship of IPP with autonomy frustration and competence frustration, and would be better combined with the above paragraph.
- In your findings, the fact that both autonomy satisfaction and frustration positively predict interaction with the opposite sex, mostly means that interaction with the opposite sex is not much affected by autonomy satisfaction and frustration. Please provide an explanation for this finding. Also, in line 307, you wrote that both autonomy satisfaction and frustration is related to more social interaction, which is not very logical, and please provide a reference if you have one.

- In line 353, it seems that it should be written that the sense of autonomy frustration is strengthened. In line 355, it seems that it should be written that hinders sense of competence satisfaction. Add in line 352, which prevents the satisfaction of relatedness.
- In line 370, it seems that it should be written autonomy satisfaction.

Experimental design

-

Validity of the findings

-

Additional comments

-

Reviewer 3 ·

Basic reporting

I would like to commend the authors for thoroughly addressing the comments raised in my previous review. The revisions made to the manuscript have significantly improved the clarity and robustness of the study. Given these improvements, I believe the manuscript is now well-prepared for publication, and I recommend it be accepted.

Experimental design

same as above

Validity of the findings

same as above

---

## Round 0.3 · Major Revisions

Although important improvements were made by the authors, some important concerns still need clarification. Please, see the attached file from the reviewer.

·

Basic reporting

Requested corrections are provided within the file.

Experimental design

-

Validity of the findings

-

---

## Round 0.4 · accepted · Accept

The authors have addressed all the reviewers' comments. Therefore, the manuscript is ready for publication.

·

Basic reporting

-

Experimental design

-

Validity of the findings

-

Additional comments

-